# Systematic Analysis of the DNA Methylase and Demethylase Gene Families in Rapeseed (*Brassica napus* L.) and Their Expression Variations After Salt and Heat stresses

**DOI:** 10.3390/ijms21030953

**Published:** 2020-01-31

**Authors:** Shihang Fan, Hongfang Liu, Jing Liu, Wei Hua, Shouming Xu, Jun Li

**Affiliations:** 1Oil Crops Research Institute of the Chinese Academy of Agricultural Sciences, Key Laboratory of Biology and Genetic Improvement of Oil Crops, Ministry of Agriculture and Rural Affairs, Wuhan 430062, China; fansh2014@sina.com (S.F.); wanghw@webmail.hzau.edu.cn (H.L.); liujing@oilcrops.cn (J.L.); huawei@oilcrops.cn (W.H.); 2Henan key laboratory of Plant Stress Biology, School of Life Sciences, Henan University, Kaifeng 475004, China

**Keywords:** rapeseed, DNA methylase, DNA demethylase, systematic analysis, salt/heat stress

## Abstract

DNA methylation is a process through which methyl groups are added to the DNA molecule, thereby modifying the activity of a DNA segment without changing the sequence. Increasing evidence has shown that DNA methylation is involved in various aspects of plant growth and development via a number of key processes including genomic imprinting and repression of transposable elements. DNA methylase and demethylase are two crucial enzymes that play significant roles in dynamically maintaining genome DNA methylation status in plants. In this work, 22 DNA methylase genes and six DNA demethylase genes were identified in rapeseed (*Brassica napus* L.) genome. These DNA methylase and DNA demethylase genes can be classified into four (*BnaCMTs*, *BnaMET1s*, *BnaDRMs* and *BnaDNMT2s)* and three (*BnaDMEs*, *BnaDML3s* and *BnaROS1s*) subfamilies, respectively. Further analysis of gene structure and conserved domains showed that each sub-class is highly conserved between rapeseed and *Arabidopsis*. Expression analysis conducted by RNA-seq as well as qRT-PCR suggested that these DNA methylation/demethylation-related genes may be involved in the heat/salt stress responses in rapeseed. Taken together, our findings may provide valuable information for future functional characterization of these two types of epigenetic regulatory enzymes in polyploid species such as rapeseed, as well as for analyzing their evolutionary relationships within the plant kingdom.

## 1. Introduction

Epigenetics is the study of heritable phenotypic changes that do not involve changes in the DNA sequence. Mounting evidence has implied DNA methylation, as one kind of epigenetic regulatory pathways, in various aspects of plant growth and development via a number of key processes including genomic imprinting and repression of transposable elements [1,2,3,4,5,6,7]. DNA methylation mainly occurs as modifications of cytosine into 5-methylcytosine (5-meC), in sequence targets such as CG, CHG, CHH (where H stands for A, C or T), and as conversion of purines into N6-methyladenine (N6-mA) and 7-methylguanine (7-mG) [8,9].

The actual DNA methylation patterns and levels are related to the function of DNA methyltransferase (DNA-MTase) and DNA demethylase (DNA-deMTase). DNA-MTase is essential for DNA methylation, and the enzymes belonging this family could be sorted into four categories based on the protein structure and function, respectively: methyltransferase 1 (MET1), chromomethylase (CMT), domains rearranged methyltransferase (DRM) and DNA methyltransferase-2 (DNMT2) [10,11]. MET1 functions to maintain the DNA methylation level of CG during DNA replication [12,13]. CMT, which is known to be unique in plants, helps to maintain DNA methylation of CHG, CHH and heterochromatin status [14,15]. DRM mediates the de novo methylation of the asymmetric site CHH sequences of DNA. De novo DNA methylation is mediated by RNA-directed DNA methylation (RdDM), a plant-specific RNA silencing pathway directed by 24-nt small interfering RNAs (siRNA) [16,17]. DNMT2, another category of methyltransferase that is conserved in many plants, can methylate tRNA(Asp)C38 in vitro [18]. To date, DNA-MTase have already been identified and characterized in several plant species, including *Arabidopsis* [19], *Oryza sativa* [20], *Daucus carota* [21], *Hordeum vulgare* [22], *Zea mays* [23] and *Lycopersicon esculentum* [24].

The precise status of DNA methylation depends not only on the function of DNA-MTase, but also on the DNA-deMTase. DNA-deMTases are contain a conserved DNA glycosilase motif and play a critical role in DNA demethylation by excision of the 5-meC in all sequence contexts (CG, CHG and CHH) [25]. Like DNA-Mtases, DNA-deMTases can be divided into four subfamilies: Repressor Of Silencing1 (ROS1), DEMETER (DME) and DEMETER-like2/3 (DML2 and DML3). Among these sub-families, ROS1 is responsible for the demethylation in the somatocyte of different plant tissues, inhibits the DNA methylation in gene promoter regions and transposons, and is involved in responses to the biotic and abiotic stresses [26,27,28], while DME is primarily expressed only in the central cell of the female gametophyte (the progenitor of the endosperm) and can influence the development of embryo and endosperm during seed development [29]. DML2/3 encode the proteins with DNA glycosylase activity that are involved in maintaining methylation marks, but their precise biological role remains unknown [30].

DNA methylation is influenced by the environmental stresses (both biotic and abiotic stresses) and may contribute to the adaptation of plants to stress [31]. Biotic stresses (e.g., pathogenic infection, virus and pathogen) can lead to two contrasting effects on the levels of methylation in plants: hypermethylation on the genome-wide level and hypomethylation of resistance-related genes [32,33]. Abiotic stresses (e.g., chilling, heat, drought, salt, planting density, heavy metal, rubbing and cutting) can generally result in decrease or increase the DNA methylation level [34,35,36,37]. 

Rapeseed (*Brassica napus* L.) is one of the most important oil crops in the world. It is a polyploid species(AACC) formed by recent allopolyploidy between the ancestors of *B. oleracea* (CC) and *B. rapa* (AA) [38]. In turn, genomes of *B. rapa* (AA) and its sister species *B. oleracea* (CC) were almost complete triplications of the genome of *A. thaliana* [39,40]. The balance of genome dosage between subgenomes is crucial for the formation of allopolyploid, and DNA methylation is no doubt playing significant roles in this process. However, systematic identification and characterization of DNA-MTase and DNA-deMTase genes in rapeseed is still lacking. Here, we first identified 22 DNA methylase and six demethylase genes in the rapeseed genome, respectively. Then, we performed numerous subsequent analyses including phylogenetic analysis, gene structure analysis, conserved domains/chromosomal location and synteny analysis. Furthermore, we analyzed the expression profiles of these genes in various tissues as well as their expression variations after heat/salt stresses treatments. Our study may provide valuable information for future functional characterization of these two kinds of epigenetic regulatory enzymes in polyploid species rapeseed as well as for analyzing their evolutionary relationships in the whole plant kingdoms.

## 2. Results

### 2.1. Identification of DNA Methyltransferases and Demethylases Genes in Rapeseed Genome

22 DNA-MTase genes and six DNA-deMTase genes were identified upon BLASTP-querying the rapeseed database with the full-length nucleic acid sequences of DNA-MTase and DNA-deMTase genes of *A. thaliana*. The newly-identified genes were named after their *A. thaliana* homologs gene (Table 1). The number of DNA-MTase and DNA-deMTase genes in *B. napus* (22 and six, respectively) were more than that of in *A. thaliana* (seven and four, respectively) and *O. sativa* (nine and three, respectively). The 22 DNA-MTase genes in rapeseed could be classified into the four subfamilies, with six *CMT* (*BnaCMTa-f*), six *MET1* (*BnaMET1a-f*), two *DNMT2* (*BnaDNMT2a-b*) and eight *DRM* (*BnaDRMa-h*) genes. The six DNA-deMTase genes belonged to three subgroups, with genes each of the *DME* (*BnaDMEa-b*), *DML3* (*BnaDMLa-b*) *ROS1* (*BnaROS1a-b*) families (Table 1). Both the A and the C genomes contribute with 14 genes to the identified repertoire. We also analyzed the gene characteristics, including coding sequence direction, start position, end position, sequence length, molecular weight (MW), isoelectric point (PI) and subcellular localization. The length of the *ORFs* from the start to stop codons varied from 1128 (*BnaDNMT2b*) to 5607 bp (*BnaDMEa*), and the encoded polypeptides ranged from 375 to 1868 amino acids (with an average length of 879 aminoacids for DNA-MTaseand 1411 for DNA-deMTase), with predicted molecular masses ranging from 42.49 to about 207.48 kDa (with an average of 99.09kDa for DNA-MTase and 158.69 kDa for DNA-deMTase). Computed PI ranges from 4.87 to 9.01 (with an average PI of 6.09 for DNA-MTase and 7.59 for DNA-deMTase) (Table 1). Most of the proteins were predicted to localize in the cell nucleus, except for BnaDRMa and BnaDRMb which were predicted to be extracellular proteins (Table 1). 

### 2.2. Multiple Sequences Alignment and Phylogenetic Analysis in Rapeseed, Arabidopsis and Rice

The identified DNA-MTase and DNA-deMTase genes in *B. napus,* together with those present *A. thaliana* and *O. sativa* genomes, were submitted to multiple sequence alignments and the corresponding phylogenetic tree was constructed (Figure 1A). Phylogenetic analysis showed that DNA-MTase from these three plant species were divided into four clusters (MET1, CMT, DRM and DNMT2). Similarly, DNA-deMTase could be separated into three sub-clades, (ROS1, DME, and *DML3*). As expected, rapeseed genes proved to be more similar to those of the dicotyledon plant (*Arabidopsis*) than to the monocotyledon plant (rice), in agreement with their evolutionary relationships. Among these rapeseed genes, six *BnaMET1s*, six *BnaCMTs*, eight *BnaDRMs*, two *BnaDNMT2s* were classified into the DNA-MTase gene family. And two *BnaDMEs*, two *BnaDML3s*, two *BnaROS1s* were classified into the DNA-deMTase gene family. No *Brassica napus* genes showed homology to the *AthDML2* sub-family. As observed also for *A. thaliana* and *O. sativa*, DNA-MTase genes are more abundant than DNA-deMTase genes. 

### 2.3. Chromosomal Location and Synteny Analysis

The twenty-eight identified genes were dispersed unevenly though seventeen chromosomes (Figure 2). Chromosome-A06 contains the most genes (*BnCMTc*, *BnCMTa*, *BnDNMT2b*, and *BnMET1a*), whereas chromsomes A3 and A8 did not contain any target gene and the other chromsomes have one or two genes. Most of the DNA-Mtase and DNA-deMTase genes are located at the start or end parts of the chromosomes.

We conducted a synteny analysis on these target genes to investigate the duplication events occurring in the rapeseed and *Arabidopsis* DNA-MTase and DNA-deMTase gene family (Figure 3). Three-way homology was observed between Bna-A05, Bna-C04 and Ath-chr2 (*BnaROS1a*, *BnaROS1b* and *AthROS1*); Bna-A10, Bna-C09 and Ath-chr5 (*BnaDMEa*, *BnaDMEb* and *AthDME*); Bna-A06, Bna-C07 and Ath-chr5 (*BnaDNMT2b*, *BnaDNMT2a* and *AthDNMT2*). Two duplication events of Ath-chr5 into Bna-A02/Bna-A10 and Bna-C02/Bna-C09 and (*BnaDRMe*, *BnaDRMf*, *BnaDRMg*, *BnaDRMh* and *AthDRM2*) were found. Two separate cases of homology were found between chromosomes Bna-A01, Bna-C01 and Ath-chr4 (*BnaCMTe*, *BnaCMTf* and *AthCMT2*; *BnaDML3a*, *BnaDML3b* and *AthDML3*), presumably evolved from longer segment duplication. *BnaCMTa/b*, *BnaCMTc/d*, *BnaMET1a/b*, *BnaMET1d/c/e/f*, *BnaDRMa/b*, *BnaDRMc/d* evolved from intra-genome gene duplication, independent of the *Arabidopsis* genome. 

### 2.4. Gene Structure and Conserved Motif Distribution Analysis

To study the component of the target genes structure, the amount and distribution of exons/introns were examined (Figure 1B). Each gene contains between nine and 24 exons, with an average number of 21 exons for *BnaCMT*, 12.5 for *BnaMET*, 10 for *BnaDNMT2* 10, 11.75 for *BnaDRM*, 21.5 for *BnaDME*, 24 for *BnaDML3* 24 and 23 exons for *BnaROS1*. The exon/intron structures of most genes in the same sub-group have a similar number of exons/intros, with small variations between zero and five. In addition, every sub-family appeared to share structural similarities with their homologous gene in *Arabidopsis*. However, some genes displayed more volatile structures and exhibited distinct intron/exon arrangements (e.g., *BnaMET1a/b*, *BnaDRMa/b/c*, and *BnaDMEa*). In our monocotyledon model plant (rice), all the homologous genes have greater variation in exon/intron number than *Arabidopsis* and rapeseed. 

The conserved motifs of every sub-family were analyzed separately by CDD web server. Nine conserved motifs were identified (Figure 1C), generally present at the C-terminus of DNA-MTase/deMTase protein sequences. The sub-family of CMTs harbour BAH, CD and MTASE functional domains. MET1s contain DNMT-RFD, BAH and Mtase domains, whereas DNMT2s only contain MTASE. DRMs carry the UBA and MTASE domains. All of the DNA-MTase proteins have the same conserved function domain of a Cytosine-C5 specific DNA methylases motif, MTASE. Especially of the members of DNMT2s, almost the entire amino acid sequence is a core conserved domain of MTASE. The three sub-families DML3s, DMEs, and ROS1s contain the same conserved domains of END03c, Perm and DNA glycoslyase (DNA glycoslyase, which is the core domain of the DNA-deMTase proteins). Remarkably, all the core conserved domains are located at the C-terminus in both DNA-Mtase and DNA-deMTase protein sequences. In general, conservation of these motifs in the target proteins was higher in rapeseed and *Arabidopsis*.

### 2.5. Expression Patterns of DNA Methylase and Demethylase Genes in Various Tissues of ZS11

To gain insights into the expression of these DNA-MTase/deMTase genes, various tissues of ‘ZS11′ (rapeseed cultivar) were analyzed. A DNA-MTase/deMTase genes are widely expressed in these 22 tissues (Figure 4), which shows that DNA-MTase/deMTase genes play a regulatory role in different stages and tissues of plant growth and development.

Different expression patterns were found for each gene. These patterns cluster in three broad sets. The set with higher transcription levels throughout the 22 tissues contained 10 genes, of which *BnaROS1a*, *BnaROS1b*, *BnaDRMh* and *BnaDRMb* show the highest expression in almost all tissues. A second set of 12 genes with moderate expression levels was found, containing *BnaMET1e* and *BnaMET1f* (which have lower transcription levels throughout except in the genital organs (pistil, silique, embryo, ovule, but not the pericarp) and *BnaDMEa* and *BnaDMEb* (whose transcription levels were especially higher in 15d-old ovule). The final, lower expression, set of genes contains six genes, including *BnaDRMf*, *BnaMET1b*, and *BnaMET1a* (whose transcripts are the least abundant in almost all tissues) and *BnaDRMd* which, in spite of being one of the least transcribed genes in most tissues, is detected in mature seeds and 35-day embryos. 

### 2.6. Correlation between Gene Expression Patterns and Methylation Level in the Endosperm 

Methylation levels (%) data obtained in our previous studies of our team [5] were then used to further explore the relationship between the degree of whole-genome methylation and the expression level of these target genes. A comprehensive horizontal comparative analysis of methylation degree and gene expression level of two varieties of *B. napus*, YN171 and 93275, 30-DAF endosperm was performed (Figure 5A). Analysis of the DNA methylation patterns by sequence location (gene regions and up/downstream regions) and type of DNA methylation sites showns that CHG and CHH are methylated more frequently in YN171 than in 93275, whereas in CG positions methylation is identical in YN171 and 93275. 

The 30-DAF endosperm transcriptome level of YN171 and 93275 (Figure 5B) shows subtle differences: *BnaDMEb* expression is slightly higher in 93275. Methylases of the *MET1* sub-family maintain the DNA methylation level of CG, and their expression level is generally higher in 93279, except for *BnaMET1a/b,* which are not expressed in either 93275 or YN171. The DNA methylation of CHG and CHH sites, in turn, largely depends on the expression of *CMTs* and *DRMs*, of which *BnaCMTb/e*, and BnaDRMf are slightly more expressed in YN171.

### 2.7. qPCR Analysis of DNA Methylase and Demethylase Genes in Response to Some Abiotic Stress

30-day seedlings were treated with 300 mM NaCl and exposed to 50 °C temperatures separately for 24 h. Real-time quantitative PCR analysis showed that the expression most rapeseed DNA methylase genes and demethylase genes were up- or down-regulated both response to hot stress and salt stress, while some gene expression levels were not significantly different (Figure 6).

In both stress conditions, *BnaCMTa* was significantly down-regulated (by 2-fold) compared to the control group, while most other members of *BnaCMTs* did not change appreciably. The members of the *BnaMETs* gene family showed different degrees of down-regulation response to salt stress, but their expression levels were not affected by the high temperature stress. *BnaDNMT2s* were approximately 2-fold up-regulated after both salt stress and high temperature stress. Among the *BnaDRMs* subfamily genes, *BnaDRMd* showed similar levels of down-regulation under salt stress, while *BnaDRMa*, *BnaDRMg* and *BnaDRMh* showed relatively high up-regulation by both stresses. Opposite responses to both stresses were only found for *BnaDRMc*, which showed a slight down-regulation response to salt stress and a 1.7-fold up-regulation response to hot stress.

*BnaDMEs*, *BnaDML3s*, and *BnaROS1s* are demethylase genes, and most of them are mildly up-regulated for salt stress and high temperature stress. Only *BnaDML3a*, *BnaROS1a* and *BnaROS1b* show some down-regulation response to salt stress, while *BnaROS1a* was also less expressed under heat stress.

## 3. Discussion

In this study, we found that the genes encoding rapeseed DNA-MTases and DNA-deMTases contain 13 introns and 22 introns on average, respectively. Similar situations were also found in other plant species including A. thaliana (with an average of 16 introns in DNA-MTase, and DNA-deMTase) and rice (with an average of 10 introns in DNA-MTase and 15 in DNA-deMTase genes). However, the average number of introns in all coding genes in rapeseed is only 3.90, 4.86 in A. thaliana is 4.86 and 3.40 in rice. Why did plant evolve so many introns in these two gene families which encode enzymes with crucial roles in mantaining DNA methylation status? It is known that introns are ubiquitous in the genome of eukaryotes, and can regulate gene expression, including transcription, RNA stability, export, and even translation through exon junction complexes [42]. The function and structure of genes evolve in different directions, which promotes the adaptation of species to changes in family genetic structure during evolution [43]. Whether the more abundant introns will lead to more alternatively spliced transcripts in rapeseed DNA-MTases and DNA-deMTases, or whether more introns provide more protection to reduce the mutation frequency of those coding genes that are vital and indispensable for plant survival remains to be addressed in future comparative genomics studies.

The expression levels of *BnaMET1e* and *BnaMET1f* in almost all tissues were low, and only in the ovule of 15 DAF were they particularly high. This observation is consistent with previous reports that the main function of MET1 is to maintain the methylation of cytosine in CG, and MET1 can affect the morphological characteristics of plants and regulate the time of flowering [44]. MET1 is widely involved in the development of morphological features in their own growth stages, the formation of gametophytes in the reproductive stage, and the stage of embryos development [45].

The expression profiles of *BnaDMEa* and *BnaDMEb* were uniform and low in the vast majority tissues, but only high expressed in the ovule of 15 DAF. Expression of DME (which is unique to dicotyledons), was first found in the central cells of the female gametophyte and in the helper cells, affecting the development of embryos and endosperm by regulating demethylation [29]. *BnaROS1a* and *BnaROS1b* were highly expressed in every tissue, especially in the embryos of 25-35 DAF, but were lower in the mature seeds. ROS1 is expressed in all tissues from both monocotyledons and dicotyledons [26], and inhibits promoter methylation [27,28].

Rapeseed performance is severely affected by high salt concentrations in soils and high temperature. Uncovering the mechanism of salt and heat tolerance in plants and finding related functional genes are therefore essential tasks for successful molecular breeding. In order to adapt to environmental challenges, plants can make adjustments in response to various abiotic and biotic stresses [36,46]. For example, after low temperature stress the methylation level of root genome decreases in maize seedlings [47]. Lack of water in peas will cause osmotic stress, leading to closure of the stomata and an increase in the degree of methylation of the genome [34]. In *Arabidopsis* subjected to salt stress hypermethylated regions were mainly distributed at the 5’ and 3’ terminal and exon regions, and the intron sites were not significantly changed [36].

This study found that the transcription level of some target genes changed significantly in response to heat or salt stress. In particular, expression of *BnaCMTa* showed extremely significant down-regulation under salt stress and heat stress. The expression of the members of the *BnaMETs* subfamily was decreased by the applied stressors and was more sensitive to salt stress than heat stress. In contrast, the *BnaDNMTs* subfamily was significantly more expressed under salt stress and heat stress. *BnaDRMs* were generally mildly down-regulated upon salt and heat treatment, with the exception of *BnaDRMa* and *BnaDRMg*, which were significantly up-regulated. The remaining demethylases (with the exception of *BnaROS1a*) were somewhat up-regulated by these stresses. Under different abiotic stresses, the expression levels of methyltransferase gene and demethylase gene changed appreciably. It is therefore possible that the methylation of some key genes (such as *BnaCMTa*, *BnaDNMTa/b*, *BnaDRMa/g*, *BnaDMEa/b*, *BnaDML3a/b*) is essential for the abiotic response of plants. The specific mechanism underlying this requires further research and analysis. Interestingly, *cis*-acting elements in the promoter sequence contain many elements related to resistance (Appendix A).

Abiotic stress can induce methylation variation in plant DNA, potentially affecting gene expression regulation and transposon activity, and therefore changing the expression of genetic information, and providing plasticity for plants to adapt to poor environments. Therefore, methylation variation has been considered as an important mechanism for plants to respond to adversity and has been widely studied. Many questions remain, including how plants perceive stress and how to activate adaptation mechanisms. Since the regulatory mechanism of cells is an extremely complex multi-factor synergy, DNA methylation regulation may be just one factor regulating gene expression.

## 4. Materials and Methods

### 4.1. Identification of DNA Methylase and Demethylase Proteins in Rapeseed

DNA methylation mechanisms in *A. thaliana* (a model dicotyledon plant) have previously been studied in depth. DNA methylase and demethylase genes in this species have been annotated in TAIR (*Arabidopsis thaliana*, dicotyledons, TAIR10, https://www.arabidopsis.org/). To identify potential members of the homologous genes in rapeseed, the amino acid sequences of *Arabidopsis* DNA methylase and demethylase were used as queries in BLASTp (E value > 10^−5^) searches against the *B. napus* annotation database version 4.1 (http://www.genoscope.cns.fr/brassicanapus/). To confirm the accuracy of these predicted genes, amino acid sequences were analyzed using CDD V. 3.15 and sequences without conserved domains were excluded. The homologous DNA methylase and demethylase proteins in *Oryza sativa* (v7_JGI, monocotyledons) were obtained from Phytozome v11 (https://phytozome.jgi.doe.gov/pz/portal.html) [48]. The ProtComp 9.0 cello web server (http://linux1.softberry.com) was used to predict the sub-cellular localization of these genes, and the ProtParam program (http://web.expasy.org/protparam/) was used to obtain the theoretical molecular weight (MW) and isoelectric point (pI) of these proteins. 

### 4.2. Multiple Sequence Alignment and Phylogenetic Analysis

All putative DNA methylase and demethylase proteins in *B. napus* were aligned with the proteins in *Arabidopsis* and *O. sativa* using MUSCLE with default parameters (version 3.8, Hinxton, Cambridge, UK) [49]. The unrooted phylogenetic tree was constructed using the full length sequences of proteins with MEGA7 [50] software using the neighbor joining (NJ) methods. The bootstrap test was performed using 1000 iterations. 

### 4.3. Chromosome Localization and Collinearity Analysis

The locations of identified genes on the chromosomes of *B. napus* were taken from the GFF (Graphics File Formats) file, which was downloaded from genome database. The collinearity pattern was analyzed using MCScanX according to the *B. napus* and *Arabidopsis* genome annotation information, to obtain the corresponding chromosome homologous segment [51]. 

### 4.4. Gene Structure and Conserved Domains Analysis

Gene structure Display Server 2.0 (http://gsds.cbi.pku.edu.cn/index.php) [52] was used to show exon/intron structure of each DNA methylase and demethylase genes. Amino acid sequences and the conserved domains of these proteins were analyzed by using conserved domain database (https://www.ncbi.nlm.nih.gov/Structure/cdd/wrpsb.cgi) [53]. In total, 2kb long upstream regions of each gene were extracted and analyzed for its *cis*-acting elements [54]. 

### 4.5. Expression Analysis of DNA Methylase and Demethylase Genes

In order to investigate the expression patterns of DNA methylase and demethylase genes, we analyzed data previously obtained by our group, including the whole genome bisulfite sequencing data and the transcriptome from the endosperm of immature seeds 30 days after flowering (DAF) from 93275, YN171 [5], and the transcriptome of 22 different tissues of ZS11 cultivars well as qPCR analysis of “ZS11” after heat/salt stress treatments. The expression profiles of the DNA methylase and demethylase genes were studied, and the transcript abundance was calculated by RPKM (Reads per Kilobase per Million Mapped Reads, value of log 2). Hierarchical clustering analysis of transcription profiles was performed using the hclust command in R. The methylation level from 2 kb upstream to 2 kb downstream of genes was analyzed to investigate the DNA methylation pattern around CG, CHG and CHH residues.

### 4.6. Plant Material and Treatments

All the rapeseed lines introduced in our studies belong to *Brassica napus* L. (semi-winter rapeseed cultivars) and were grown in the fields at Wuhan, China. Endosperms of YN171 and 93275 were collected from 30DAF immature seeds, for both RNA-seq and whole genome bisulfite sequencing. 22 different tissues of ZS11 (7d seeding, root and leaf of trefoil stage, root and leaf of bolting stage, base stem, top stem, axillary bud, flower bud, petal, stamen, pistil, rosette leaf of flowering, cauline leaf of flowering, silique of 3DAF and 10DAF, silique wall of 15DAF and 25DAF, ovule of 15DAF, embryo of 25DAF and 35DAF, mature seed.) were obtained. To explore the effects of abiotic stress on these genes, ZS11 was treated with 50 °C 200 mM NaCl for 24 h at the stage of 5 leaves. Each sample was collected from at least three individual plants and with two biological replicates. All the collected tissues were immediately frozen in liquid nitrogen and stored at −80 °C.

### 4.7. RNA Extraction and Quantitative Real-Time PCR

Total RNA was extracted with MiniBEST Plant RNA Extraction Kit (TaKaRa, Dalian, China). Synthesis of first-strand cDNA and RT-PCR was performed using the PrimeScript™ RT reagent Kit (TaKaRa, Dalian, China) with gDNA Eraser. qRT-PCR was conducted by Hieff^®^ qPCR SYBR Green Master Mix (YEASEN, Shanghai, China). Three replicates were performed for the analysis of each gene. The rapeseed *BnTMA7* gene was used as an internal control. The primers used in gene expression analysis are listed in Appendix A.

## Figures and Tables

**Figure 1 ijms-21-00953-f001:**
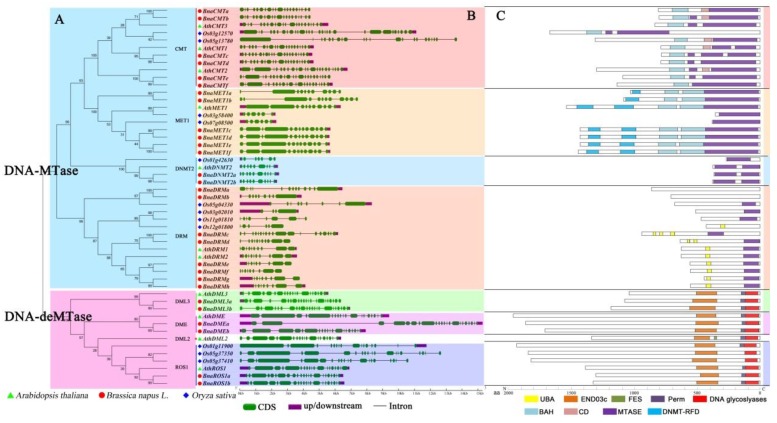
Evolutionary relationship, gene structure, and protein domain analysis of rapeseed DNA-MTase and DNA-deMTase gene families. (**A**): Family phylogenetic tree analysis. The evolutionary distances were computed using the Poisson correction method and are in units of “number of amino acid substitutions per site”. The analysis involved 51 amino acid sequences. All positions containing gaps and missing data were disregarded; (**B**): Gene structure. Exons are shown as green boxes, and introns are shown as black lines, upstream/downstream are shown as purple boxes. Some genes lack the annotation of upstream/downstream zone; (**C**): Protein domains. Schematic representation of conserved motifs. Colored boxes indicate different conserved motifs.

**Figure 2 ijms-21-00953-f002:**
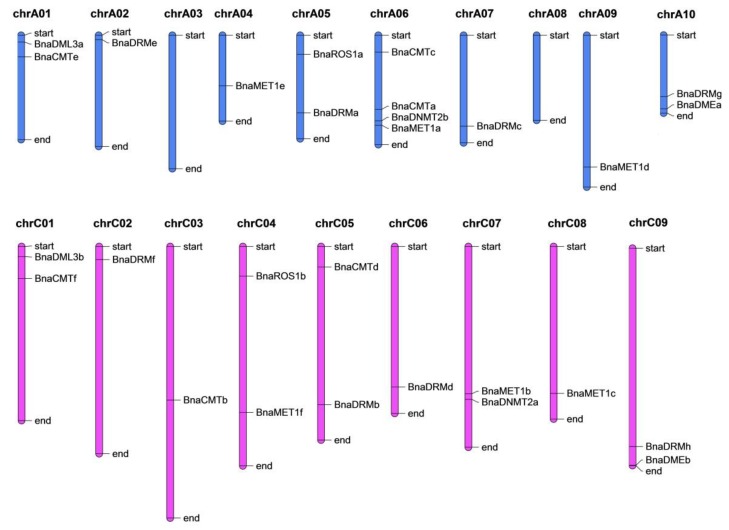
Genomic distribution of rapeseed DNA-MTase and DNA-deMTase families’ genes on chromosomes. The vertical bars with varying lengths represent *B. napus* chromosomes; short black horizontal lines indicate the position of each gene, chrA01 to chrA10 and chrC01 to chrC09 are the ten and nine chromosomes in the A_n_ and C_n_ subgenomes in *B. napus*, respectively.

**Figure 3 ijms-21-00953-f003:**
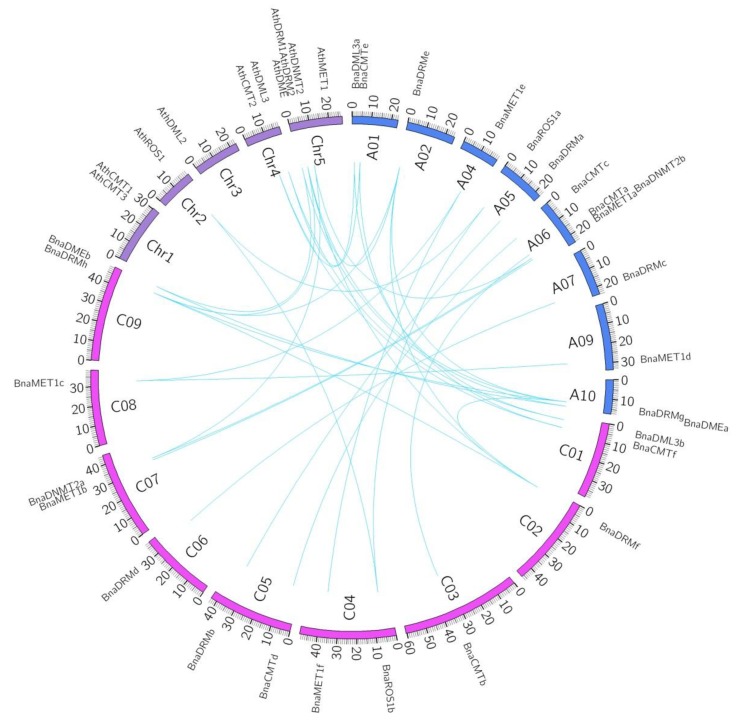
Synteny analyses of rapeseed DNA-MTase and DNA-deMTase families’ genes between *B. napus* and *Arabidopsis*. A01 to A10 and C01 to C09 represent the ten and nine chromosomes in the A_n_ and C_n_ subgenomes in *B. napus*, respectively. Chr1 to Chr5 represent the five chromosomes in *Arabidopsis*. Syntenic regions between *B. napus* and *Arabidopsis* are represented by blue lines.

**Figure 4 ijms-21-00953-f004:**
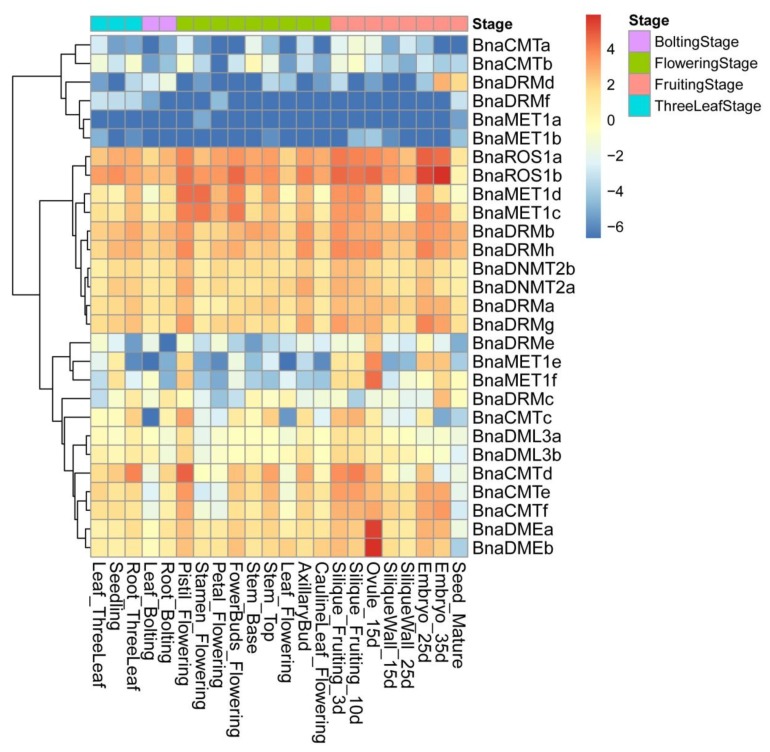
Heat map of rapeseed DNA-MTase and DNA-deMTase families’ genes expression patterns across tissues. Color scale bar at the top of each heat map represents log_2_-transformed FPKM (fragments per kilobase of exon per million fragments mapped) values for each gene, with warmer colors denoting higher expression. Tissues are divided into four periods according to the growth process: three leaf stage, bolting stage, flowering stage, and fruiting stage.

**Figure 5 ijms-21-00953-f005:**
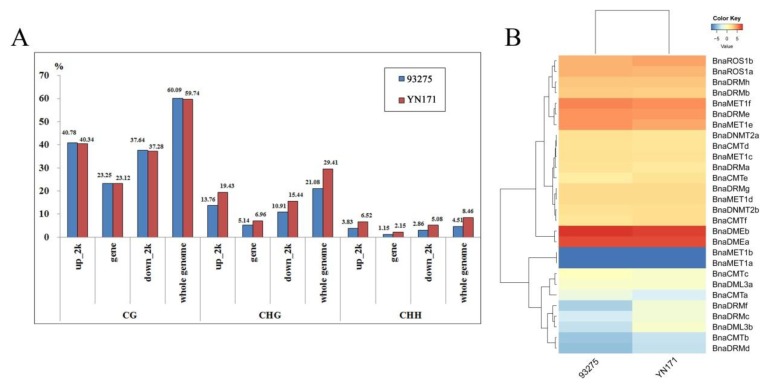
DNA methylation levels and gene expression patterns of 30 DAF endosperm in 93275 and YN171. (**A**) The whole genome DNA methylation levels (%); (**B**) heat map of rapeseed DNA methyltransferases and demethylase families gene expression patterns, color scale bar at the top of each heat map represents log_2-_transformed fragments per kilobase of exon per million fragments mapped (FPKM) values of each gene, with warmer colors denoting higher expression.

**Figure 6 ijms-21-00953-f006:**
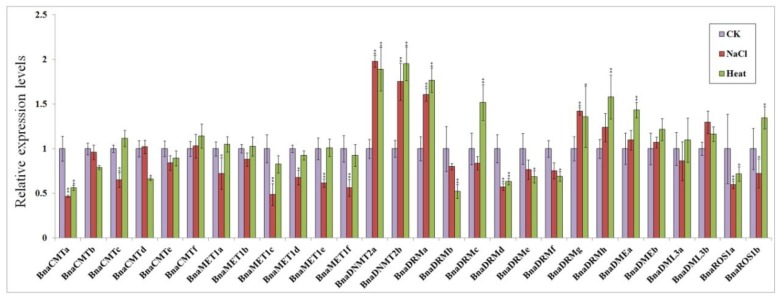
Expression patterns of target genes in response to salt and high temperature stress conditions. Three replicates were performed for the analysis of each condition: untreated control (purple), 24 h 300 mM NaCl stress (red), and 24 h 50 °C heat stress (green). Transcript levels were analyzed by quantitative real-time PCR using BnTMA7 gene as an internal control [41]. Statistically significant differences (t-test) vs. control group (CK) are indicated by asterisks: * *p* < 0.05, ** *p* < 0.01.

**Table 1 ijms-21-00953-t001:** Basic characteristic of genes encoding DNA methyltransferase (DNA-MTase) and DNA demethylase (DNA-deMTase) in rapeseed.

Gene ID	GENE NAME	Sub Cellular	Direction	Start Pos	End Pos	DNA Length	CDS Length	Exon No.	AA Length	Mol.Wt. (kD)	PI
*BnaA06g23880D*	*BnaCMTa*	Nuclear	+	16,520,582	16,525,017	4436	2418	20	805	91.94	6.57
*BnaC03g49350D*	*BnaCMTb*	Nuclear	-	34,264,132	34,268,583	4452	2409	20	802	91.61	7.52
*BnaA06g07080D*	*BnaCMTc*	Nuclear	-	3,771,918	3,776,153	4563	2367	21	788	88.78	5.15
*BnaC05g08730D*	*BnaCMTd*	Nuclear	-	4,569,531	4,573,838	4641	2391	20	796	89.4	5.06
*BnaA01g09820D*	*BnaCMTe*	Nuclear	+	4,828,328	4,834,033	5706	3297	23	1,098	122.34	6.02
*BnaC01g11520D*	*BnaCMTf*	Nuclear	+	7,147,733	7,153,381	5843	3435	22	1,144	127.61	5.7
*BnaA06g29420D*	*BnaMET1a*	Nuclear	-	20,057,862	20,064,228	6367	3099	11	1032	117.19	9.01
*BnaC07g27300D*	*BnaMET1b*	Nuclear	+	32,855,407	32,862,845	7439	3297	12	1098	125.27	8.82
*BnaC08g34700D*	*BnaMET1c*	Nuclear	+	32,763,560	32,769,056	5705	4296	13	1431	162.1	5.92
*BnaA09g42250D*	*BnaMET1d*	Nuclear	+	29,386,022	29,391,515	5642	4299	13	1432	162.15	6.07
*BnaA04g13390D*	*BnaMET1e*	Nuclear	-	11,296,866	11,302,393	5666	4299	13	1432	161.45	6.05
*BnaC04g35500D*	*BnaMET1f*	Nuclear	-	37,010,080	37,015,638	5700	4326	13	1441	162.14	6.07
*BnaC07g29260D*	*BnaDNMT2a*	Nuclear	+	34,107,074	34,109,326	2468	1164	10	387	43.97	5.49
*BnaA06g27730D*	*BnaDNMT2b*	Nuclear	-	19,065,967	19,068,137	2342	1128	10	375	42.49	5.68
*BnaA05g22820D*	*BnaDRMa*	Extracellular	+	17,302,556	17,308,756	6464	2655	14	884	96.78	5.01
*BnaC05g36090D*	*BnaDRMb*	Extracellular	+	35,279,390	35,282,899	3889	2124	9	707	78.88	4.87
*BnaA07g28030D*	*BnaDRMc*	Nuclear	+	20,274,800	20,280,859	6187	2823	19	940	106.79	6.47
*BnaC06g30710D*	*BnaDRMd*	Nuclear	+	31,350,535	31,353,717	3183	1911	11	636	72.2	5.1
*BnaA02g02230D*	*BnaDRMe*	Nuclear	-	970,840	973,703	3262	1677	10	558	62.91	5.31
*BnaC02g05620D*	*BnaDRMf*	Nuclear	-	2,918,874	2,921,525	2652	1668	9	555	62.39	5.17
*BnaA10g19200D*	*BnaDRMg*	Nuclear	+	13,731,342	13,734,360	3780	1671	11	441	49.53	7.01
*BnaC09g42890D*	*BnaDRMh*	Nuclear	+	44,252,075	44,255,166	4126	1671	11	556	62.09	5.26
*BnaA10g25630D*	*BnaDMEa*	Nuclear	-	16,440,585	16,454,849	15,293	5607	23	1,868	207.48	8.3
*BnaC09g50670D*	*BnaDMEb*	Nuclear	+	48,435,123	48,442,674	7929	5133	20	1,710	189.21	7.98
*BnaA01g03070D*	*BnaDML3a*	Nuclear	-	1,479,536	1,485,590	6369	3243	24	1080	124.52	8.59
*BnaC01g04300D*	*BnaDML3b*	Nuclear	-	2,253,312	2,260,155	6948	3549	24	1182	135.52	8.37
*BnaA05g07800D*	*BnaROS1a*	Nuclear	+	4,240,657	4,246,594	6512	3903	24	1300	146.68	6.27
*BnaC04g08810D*	*BnaROS1b*	Nuclear	+	6,603,988	6,609,839	6583	3975	22	1324	148.73	6.08

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
