# Peer review of "Systematic Analysis of the DNA Methylase and Demethylase Gene Families in Rapeseed (Brassica napus L.) and Their Expression Variations After Salt and Heat stresses"

_ijms, 2020, doi:10.3390/ijms21030953_

Round 1
Reviewer 1 Report
The manuscript by Fan and colleagues describes research aimed to identify DNA methyltransferases and demethylases in rapeseed. The topic is very interesting for a plant research community and for a broader audience involved in epigenetic research, but the quality of the provided manuscript doesn't allow me to support its publication.
Below I list only a few examples of issues, which I found in the manuscript. One may continue the list.
1) The manuscript requires extensive editing of English, there are lots of grammatical mistakes, many uncompleted sentences, etc.
2) The introduction has conceptual mistakes: "Epigenetics is the study of stable genetic modifications..."
"In plants, DNA methylation mainly occur at the sequence of 5-methylcytosine (5-meC), like CG, CHG, CHH (H stands for A, C or T), but animals with the N6-mA, 7-mG primarily."
3) Figure legends are scarce; they should contain more details explaining the presented data.
4) A paragraph title "2.6. Correlation between gene expression patterns and methylation level in the endosperm" doesn't reflect the content of it. No correlation analysis is shown.
5) In several places conclusions are not supported by experimental data, for example, Fig 5B: "The heat map showed that the transcriptome level of YN171 and 93275 (Figure.5-B). The difference between them is that BnaDMEa/b are expressed more higher in 93275." According to the heat map, these two genes have similar expression levels in both variants. Fig 5A: "The DNA methylation patterns and levels showed as figure.5-A, no matter the gene sequences or the up-2k and down-2k sequences, the DNA methylation levels of CHG, and CHH are higher in YN171, but slightly lower at CG.”
There is no difference between the strains at the CG sites, and experimental repeats are required to draw any conclusion.
6) Fig. 6. What do the single and double stars and error bars mean? Error bars of several compared samples do overlap but are labeled with the double star. Are they significantly different?
Author Response
Reviewer #1:
The manuscript by Fan and colleagues describes research aimed to identify DNA
methyltransferases and demethylases in rapeseed. The topic is very interesting for a plant research community and for a br oader audience involved in epigenetic research, but the quality of the provided manuscript doesn't allow me to support its publication.
Below I list only a few examples of issues, which I found in the manuscript. One may continue the list.
1) The manuscript requires extensive editing of English, there are lots of grammatical mistakesmany uncompleted sentences, etc.
Response Following your good suggestion , t he language of this manuscript has been smoothed by EditSprings (https://www.editsprings.com)
2) The introduction has conceptual mistakes: "Epigenetics is the study of stable genetic modifications..."
"In plants, DNA methylation mainly occur at the sequence of 5 methylcytosine (5 meC), like CG, CHG, CHH (H stands for A, C or T), but animals with the N6 mA, 7 mG primarily."
Response: We have revised the conceptual mistakes throughout the whole manuscript.
3) Figure legends are scarce; they should contain more details explaining the presented data.
Response: The Figure legends have been revised to be more concrete according to your good suggestion.
4) A paragraph title "2.6. Correlation between gene expression patterns and methylation level in the endosperm" doesn't reflect the content of it. No correlation analysis is shown.
Response: Than ks for reminding us. And we have made some modifications in this paragraph in the revised version.
5) In several places conclusions are not supported by experimental data, for example, Fig 5B: "The heat map showed that the transcriptome level of YN171 and 93275 (Figure.5 B). The difference between them is that BnaDMEa/b are expressed more higher in 93275." According to the heat map, these two genes have similar expression levels in both variants . Fig 5A: "The DNA methylation patterns and levels showed as figure.5 A, no matter the gene sequences or the up 2k and down 2k sequences, the DNA methylation levels of CHG, and CHH are higher in YN171, but slightly lower at CG.”
There is no difference between the strains at the CG site s , and experimental repeats are required to draw any conclusion.
Response: Thanks for pointing out these mistakes. We have corrected them in the revised version .
6) Fig. 6. What do the single and double stars and error bars mean? Error bars of several compared samples do overlap but are labeled with the double star. Are they significantly different?
Response: We have added some Figure legend s. Figure 6. Expression patterns of target genes in response to salt and high temperature stress conditions. Three replicates were performed for the analysis of each condition: untreated control (purple), 24 hours 300 mM NaCl stress (red), and 24 hours 50 °C heat stress (green). Transcript levels were analyzed by quantitative real time PCR using BnTMA7 gene as an internal control [43]. Statistically significant differences (t test) vs. control group (CK) are indicated by asterisks: * p < 0.05, ** p < 0.01.)
Reviewer 2 Report
The manuscript presented by Shihang Fan , Hongfang Liu , Jing Liu , Wei Hua, Shouming Xu and Jun Li entitled `Systematic Analysis of the DNA Methylase and Demethylase Gene Families in Rapeseed (Brassica napus L.) and Their Expression Variations After Salt and Hot stresses`, presents the observation of DNA methyltransferases (MTase) and demethylases (deMTase) genes in Brassica napus and provide some data regarding the expression patterns of mentioned genes.
Generally, I found this manuscript hard to read and understand all the data presented, because of the bad way of representing and neglectful style of writing. Manuscript is poorly written, and it is should be greatly revised and completely rewritten to be published.
The current study presented by authors is a continuation of their work presented in previous publication:
Liu, J.; Li, J.; Liu, H.-f.; Fan, S.-h.; Singh, S.; Zhou, X.-R.; Hu, Z.-y.; Wang, H.-z.; Hua, W. Genome-wide screening and analysis of imprinted genes in rapeseed (Brassica napus L.) endosperm. DNA Research 2018, 25, 629-640.
But, in my opinion current is written much worse. However, manuscript parts concerning identification, sequencing and localization analysis of DNA MTase and deMTase genes is performed well in respect of scientific point of view.
Recommendation: Major Revisions
English language writing is the really serious problem. In some cases text is barely understandable. It should be totally rewritten by native speaker. Analysis of DNA methylation. In current article I found no information on how the level of cytosine methylation was measured. At this point authors referred to Liu et al., 2018 and that is all. However, I would like to mention that if authors present this manuscript as original study, the information on how bisulphite sequencing was performed should be introduced. Figure 5 (A). Analysis of DNA methylation. As a reader I am wondering if any statistical analysis were made regarding the data presented? Was the experiment repeated, how many replicates? There are no error bars, and no sign of statistical significance of observed differences in the methylation level, so is this data valuable? Discussion. The main problems of the current variant of Discussion, and the manuscript idea in total are the authors’ assumptions that the expression of certain genes involved in response to various stresses is: 1) Regulated only by cytosine methylation; 2) MTases and deMTases are choosing their target genes by themselves. However, 1) The plant cell is very complicated system consisting of different mechanisms, and the synergy of different mechanisms leads to the certain results. And, being a very important, DNA methylation is not the only factor regulating gene expression. 2) Many times MTases were shown to execute their functions as a part of protein complexes, and authors even show the evidence analyzing the conserved domains. Moreover, deMTases mostly are specific components of DNA repair machinery, and not really execute their functions in “solo” manner. Summurizing, MTases and deMTases are the “instruments” and their targeting to certain gene loci in most cases is not depended on them.
Author Response
Reviewer #2:
1) English language writing is the really serious problem. In some cases text is barely understandable. It should be totally rewritten by native speaker.
Response: Thanks for your comments or suggestions. T he language of this manuscript has been polished by EditSprings (https://www.editsprings.com/)
2) Analysis of DNA methylation. In current article I found no information on how the level of cytosine methylation was measured. At this point authors referred to Liu et al., 2018 and that is all. However, I would like to mention that if authors present this manuscript as original study, the information on how bisulphite
sequencing was performed should be introduced. Figure 5 (A). Analysis of DNA methylation. As a reader I am wondering if any statistical analysis were made regarding the data presented? Was the experiment repeated, how man y replicates? There are no error bars, and no sign of statistical significance of observed differences in the methylation level, so is this data valuable?
Response: The data of methylation level frequence (%) and transcriptome level was used from previous studies in our team, used the same sample as data sources. The method of identification of Methylated Cytosines as follows:
To test whether each cytosine (covered by at least four reads) was methylated, the proportion of methylated reads to unmethylated r eads (calculated by Bismark's methylation extractor script) was compared to the background error rate by using a binomial test. The background false positive error rate (sequencing errors + conversion errors) was calculated by using reads mapping to the un methylated chloroplast genome. The number of methylated cytosines was calculated independently for each library. Correction for multiple testing was performed with Storey’s q values (Storey and Tibshirani, 2003) with an FDR of 0.05. So there is no detail information about on how bisulphite sequencing was performed, and no error bars bars on the histogram of Figure 5 A.
3) Discussion. The main problems of the current variant of Discussion, and the manuscript idea in total are the authors’ assumptions that the expression of certain genes involved in response to various stresses is: 1) Regulated only by cytosine methylation; 2) MTases and deMTases are choosing their target
genes by themselves. However, 1) The plant cell is very complicated system consisting genes by themselves. However, 1) The plant cell is very complicated system consisting of of different mechanisms, and the synergy of different mechanisms leads to the certain results. And, different mechanisms, and the synergy of different mechanisms leads to the certain results. And, being a very important, DNA methylation is not the only factor regulating gene expression. 2) being a very important, DNA methylation is not the only factor regulating gene expression. 2) Many times MTases were shown to execute their functions as a pMany times MTases were shown to execute their functions as a part of protein complexes, and art of protein complexes, and authors even show the evidence analyzing the conserved domains. Moreover, deMTases mostly authors even show the evidence analyzing the conserved domains. Moreover, deMTases mostly are specific components of DNA repair machinery, and not really execute their functions in “solo” are specific components of DNA repair machinery, and not really execute their functions in “solo” manner. Summurizing, MTases and deMTasmanner. Summurizing, MTases and deMTases are the “instruments” and their targeting to certain es are the “instruments” and their targeting to certain gene loci in most cases is not depended on them. gene loci in most cases is not depended on them.
Response:Your suggestionsuggestion on the discussion section is very valuableon the discussion section is very valuable. And this part is rewritten. And this part is rewritten..in in the revised manuscprit.the revised manuscprit.
Kind regards,
Round 2
Reviewer 1 Report
Language has been significantly improved, now one can read the manuscript. Authors have implemented some changes but there are still multiple issues that do not allow me to support publication of this work in its current state.
Major issues:
Some statements are not supported by experimental observations.
Chapter 2.6. Fig 5A It is fair to say that the methylation of CG sites is identical.
Chapter 2.6. Fig 5B, Expression of BnaDMEa is identical in both strains.
Chapter 2.7. BnaCMTa is downregulated by more than two-fold. T-test results of multiple data sets are in doubt. Please check them once again.
Minor corrections:
Fig 5A legend. “The whole genome DNA methylation level frequence(%);” should be “The whole genome DNA methylation levels (%);”
Page 1: “and as conversion of purines into N6-methylpurine (N6-mA)” should be
“and as conversion of purines into N6-methyladenine (N6-mA)”
Page 3: “genes of A. thaliana .” correct to “genes of A. thaliana.”
Page 3: “to about 207.48 kDa(with” correct to “to about 207.48 kDa (with”
Page 3: “from 4.87 to 9.01(with” correct to “from 4.87 to 9.01 (with”
Page 7: “gene duplication , independent” correct to “gene duplication, independent”
Page 8: “wheras DNMT2s only” correct to “whereas DNMT2s only”
Page 8: Correct the sentence “The three sub-families DML3s, DMEs, and ROS1s contain the same conserved domains of END03c, Perm and DNA glycoslyase. (which is the is the core domain of the DNA-deMTase proteins).”
Reviewer 2 Report
Accept
Round 3
Reviewer 1 Report
Authors have clarified all raised issues, the manuscript is recommended for publication.